# Self-Supervision Revives Simple Multiple Instance Classification Methods in Pathology

**Ali Mammadov**[1,2]                                        ALI.MAMMADOV@IP-PARIS.FR

**Loïc Le Folgoc**[1]                                       LOIC.LEFOLGOC@TELECOM-PARIS.FR

**Julien Adam**[2]                                                          JADAM@GHPSJ.FR

**Anne Buronfosse** [2]                                          ABURONFOSSE@GHPSJ.FR

**Gilles Hayem**[2]                                                        GHAYEM@GHPSJ.FR

**Guillaume Hocquet**[2]                                              GHOCQUET@GHPSJ.FR

**Pietro Gori**[1]                                          PIETRO.GORI@TELECOM-PARIS.FR

[1] *LTCI, Telecom Paris, Institut Polytechnique de Paris*

[2] *Paris Saint-Joseph Hospital*

**Editors:** Accepted for publication at MIDL 2024

## Abstract

Multiple Instance Learning (MIL) is the current solution for classifying whole slide pathology images (WSI). MIL divides WSIs into patches, treating each slide as a bag of instances with a global label. There are two main MIL approaches: instance-based and embedding-based. The former classifies patches independently and aggregates scores for bag label prediction, while the latter performs bag classification after aggregating patch embeddings. Even if instance-based methods are more interpretable, embedding-based MILs have been preferred in the past, due to their robustness to poor feature extractors. In parallel, many works started to use self-supervised learning (SSL) for training better encoders. However, despite the use of SSL feature extractors, many works continue to endorse the superiority of embedding-based MILs. Here, we show that with a good SSL feature extractor, simple instance-based MILs, with very few parameters, obtain similar or better performance than complex, state-of-the-art embedding-based MIL methods.

**Keywords:** Self-Supervised Learning, Multiple Instance Learning, Digital Pathology

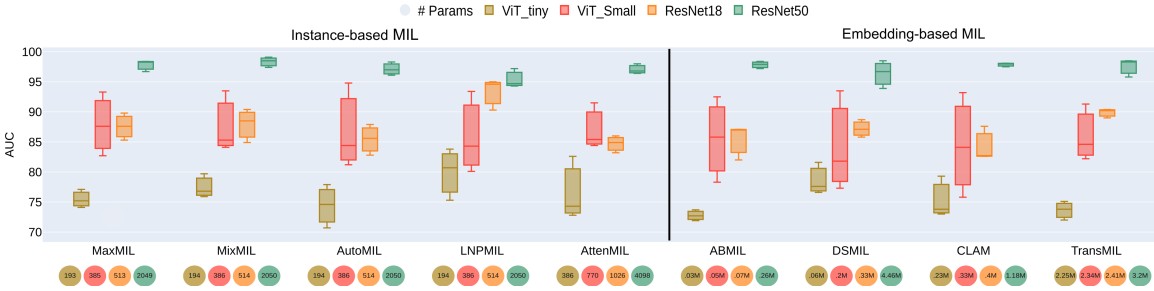

Figure 1: AUC scores on Camelyon16 dataset. Each score is an averaged result of 3 self-supervised pre-trainings: DINO, MOCO-V3 and Barlow Twins.

## Introduction

Whole Slide histopathology Image (WSI) analysis is the main tool for disease diagnosis in digital pathology. However, the gigapixel size of WSIs makes the manual analysis very time-consuming and presents significant challenges for conventional Deep Learning (DL) methods. To tackle this challenge, Multiple-Instance Learning (MIL) methods, coupled with Deep Learning Feature Extraction (FE), have emerged for WSI classification using only slide (i.e., weak) labels. Under MIL formulation, each WSI is treated as a "bag" containing multiple instances in the form of patches. The bag is labeled positive (i.e., diseased) if at least one of its patches is positive, or negative if all patches are negatives (Ilse et al., 2018). In general, the existing methodologies follow a two-step pipeline: 1) feature extraction from individual patches, and 2) MIL aggregation through a pooling operation to predict the slide label (Lu et al., 2021; Li et al., 2021). Many works in the literature focus on the second part of the pipeline, developing various aggregation methods that can be categorized into two groups: *instance-based* and *embedding-based* methods. Instance-based methods use an instance-level classifier, which predicts a score for each patch. Then, these scores are aggregated via a MIL-based pooling operator to make the final prediction for the entire slide. Common pooling operators include: average-pooling (*MeanMIL*), max-pooling (*MaxMIL*) and attention-pooling (*AttenMIL*) (Wang et al., 2019). However, these methods were only used in a few and early works in the field of Digital Pathology, such as (Campanella et al., 2019). Even if these methods are naturally interpretable and easily explainable, they highly depend on the quality of the embedding. To increase reliability, researchers proposed to aggregate features instead than scores, moving the classification head after the pooling. These are called embedding-based methods, which use deep self-attention mechanism (Ilse et al., 2018) or transformer (Shao et al., 2021), for instance, for feature aggregation. These pooling mechanisms are usually more complex (more parameters) than instance-based ones. On the one hand, this means that the model is better at feature aggregation and thus might be more accurate. On the other hand, interpretability and explainability can decrease and at the same time, computational complexity and overfitting might increase.

The lack of large, weakly annotated WSI datasets led most of the researchers to use ImageNet pre-trained models to extract features. However, as shown in (Matsoukas et al., 2022; Raghu et al., 2019), these models might not be optimal for histopathology images, due to the domain gap between natural and medical images. This motivated exploring self-supervised feature extraction methods for WSI classification and might also explain why most of the early, ImageNet-based works found that embedding-based MIL models outperformed instance-based ones (Kanavati et al., 2020; Shao et al., 2021). In digital pathology, Self Supervised Learning (SSL) has been actively used by recent studies, notably in (Li et al., 2021; Chen and Krishnan, 2021; Wang et al., 2021). All of these approaches have shown that self-supervision outperforms the ImageNet pre-training on all downstream tasks. Even if recent works provide very insightful results for WSI classification using SSL methods, they all only prove the superiority of SSL-based encoders over TL ones. In contrast, our work shows that recent SSL methods not only outperform ImageNet pre-training but also suppress the need for complex and over-parametrized embedding-based MILs. Here, we show that simple and interpretable instance-based MIL methods, when combined with robust SSL, outperform or are on par with complex embedding-based MILs, see Fig 1.

| Arch. | Pretrain. | MaxMIL | MixMIL | AutoMIL | LNPMIL | AttenMIL | ABMIL | DSMIL | CLAM | TransMIL |
|---|---|---|---|---|---|---|---|---|---|---|
| | # Params. | (193) | (194) | (194) | (194) | (386) | (0.03M) | (0.06M) | (0.23M) | (2.25M) |
| ViT-Tiny | *ImageNet* | 69.2 | 72.5 | 71.6 | 71.5 | 73.9 | 65.5 | 70.5 | 71.2 | 66.7 |
| | Barlow T. | 74.1 | 79.7 | 77.9 | **83.8** | 82.6 | 71.9 | 77.6 | 79.3 | 75.1 |
| | MOCOv3 | 75.2 | 75.9 | 70.7 | 80.7 | 74.3 | 72.7 | 76.6 | 73.8 | 72.0 |
| | SimCLR | 76.0 | 74.2 | 77.4 | 73.8 | 75.4 | 76.9 | 72.8 | 77.5 | 72.7 |
| | BYOL | 72.1 | 73.7 | 73.8 | 82.6 | 77.4 | 74.4 | 74.5 | 69.2 | 78.5 |
| | DINO | 77.1 | 76.8 | 74.6 | 75.3 | 72.8 | 73.7 | 81.6 | 73.0 | 73.8 |
| | MAE | 57.4 | 59.5 | 54.2 | 61.9 | 66.9 | 72.1 | 67.4 | 64.7 | 65.2 |
| | *Avg.* | 72.0±6.7 | 73.3±6.5 | 71.4±8.1 | **76.4**±7.4 | 74.9±4.7 | 73.6±1.7 | 75.1±4.4 | 72.9±4.9 | 72.9±4.0 |
| | # Params. | (385) | (386) | (386) | (386) | (770) | (0.05M) | (0.2M) | (0.33M) | (2.34M) |
| ViT-Small | *ImageNet* | 64.8 | 77.2 | 75.4 | 75.0 | 72.6 | 76.1 | 74.9 | 71.8 | 73.6 |
| | Barlow T. | 87.6 | 85.3 | 81.2 | 84.3 | 85.4 | 78.3 | 77.3 | 84.1 | 84.6 |
| | MOCOv3 | 93.3 | 93.5 | 94.8 | 93.4 | 91.5 | 92.5 | 93.5 | 93.2 | 91.3 |
| | SimCLR | 73.2 | 72.5 | 67.5 | 76.9 | 63.5 | 75.5 | 76.1 | 69.2 | 69.0 |
| | BYOL | 85.9 | 89.0 | 83.3 | 90.8 | 80.1 | 84.4 | 86.4 | 86.8 | 84.8 |
| | DINO | 82.7 | 84.1 | 84.4 | 80.1 | 84.4 | 85.8 | 81.8 | 75.8 | 82.2 |
| | MAE | 70.1 | 72.3 | 68.4 | 60.6 | 74.5 | 79.0 | 77.3 | 73.8 | 78.4 |
| | *Avg.* | 82.1±8.1 | **82.8**±7.9 | 79.9±9.5 | 81.0±11.0 | 79.9±9.0 | 82.6±5.7 | 82.1±6.2 | 80.5±8.3 | 81.7±6.9 |
| | # Params. | (513) | (514) | (514) | (514) | (1026) | (0.07M) | (0.33M) | (0.4M) | (2.41M) |
| ResNet18 | *ImageNet* | 66.8 | 73.4 | 70.5 | 67.8 | 72.4 | 72.9 | 60.1 | 62.5 | 70.2 |
| | Barlow T. | 89.8 | 90.4 | 85.6 | 94.6 | 86.0 | 87.1 | 85.8 | 87.6 | 90.4 |
| | MOCOv3 | 87.6 | 88.5 | 87.9 | **96.0** | 84.9 | 87.0 | 88.7 | 82.6 | 90.2 |
| | SimCLR | 89.9 | 79.2 | 78.8 | 84.8 | 76.5 | 80.1 | 86.8 | 78.8 | 91.0 |
| | BYOL | 73.6 | 74.7 | 77.4 | 74.4 | 75.5 | 74.1 | 73.8 | 75.2 | 75.1 |
| | DINO | 85.3 | 84.9 | 82.8 | 90.3 | 83.2 | 82.0 | 87.1 | 82.7 | 89.0 |
| | *Avg.* | 85.2±6.1 | 83.5±5.8 | 82.5±4.0 | **88.0**±7.8 | 81.2±4.4 | 82.1±4.8 | 84.4±5.4 | 81.4±4.2 | 87.1±6.1 |
| | # Params. | (2049) | (2050) | (2050) | (2050) | (4098) | (0.26M) | (4.46M) | (1.18M) | (3.2M) |
| ResNet50 | *ImageNet* | 64.4 | 83.2 | 77.7 | 77.1 | 72.0 | 77.2 | 80.7 | 86.3 | 78.9 |
| | Barlow T. | 98.3 | 98.5 | 96.1 | 97.2 | 96.4 | 98.4 | 98.5 | 98.1 | 98.3 |
| | MOCOv3 | 96.7 | 97.4 | 97.0 | 94.3 | 96.8 | 97.2 | 96.7 | 97.5 | 95.8 |
| | SimCLR | 81.4 | 92.3 | 92.4 | 84.5 | 91.2 | 89.9 | 88.0 | 92.4 | 92.0 |
| | BYOL | 96.4 | 97.1 | 96.0 | 90.9 | 94.4 | 95.9 | 94.5 | 97.9 | 97.9 |
| | DINO | 98.4 | **99.1** | 98.3 | 94.7 | 98.0 | 97.9 | 93.9 | 98.0 | 98.5 |
| | *Avg.* | 94.2±6.5 | **96.9**±2.4 | 96.0±2.0 | 92.3±4.4 | 95.4±2.4 | 95.9±3.1 | 94.3±3.6 | 96.8±2.2 | 96.5±2.4 |

Table 1: Results on Camelyon16 dataset. The first and second-best AUC scores in bold and underlined respectively. The best average AUC scores (per MIL) in red.

## Method

We include in our study ImageNet pre-training as baseline and 6 different SSL methods: SimCLR (Chen et al., 2020), MoCoV3 (Chen et al., 2021), MAE (He et al., 2022), DINO (Caron et al., 2021), BYOL (Grill et al., 2020) and Barlow Twins (Zbontar et al., 2021). We experiment with four backbones: *ResNet18*, *ResNet50*, *ViT-Tiny* and *ViT-Small*. All pre-trainings are done using the *solo-learn* (Costa et al., 2022). We compare 6 instance-based MIL methods (MaxMIL, MeanMIL, MixMIL (Lee et al., 2016), AutoMIL (McFee et al., 2018), LNPMIL (Gulcehre et al., 2014), and AttenMIL (Wang et al., 2019) ) with 4 SOTA embedding-based methods (ABMIL (Ilse et al., 2018), CLAM (Lu et al., 2021), DSMIL (Li et al., 2021), and TransMIL (Shao et al., 2021)).

**Results.** Tables 1 presents results for ViT-Tiny, ViT-Small, and ResNet18 backbones at x10 magnification, while for ResNet50 at x20 magnification. Across all setups, instance-based MIL methods are on par with embedding-based MIL methods (on average over the SSL method), despite having fewer parameters by several orders of magnitude. All the best-performing combinations of SSL and MIL use instance-based MIL. Notably, the simple MixMIL method, with a ResNet50 backbone pretrained with DINO on Camelyon16 at x20 magnification, achieves a new SOTA result with a 99.1 AUC score.

**Acknowledgments.** This work was performed using HPC resources from GENCI-IDRIS (Grant 2023-AD011013982R1)

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
