# OpenReview forum: "Self-Supervision Revives Simple Multiple Instance Classification Methods in Pathology"
_MIDL.io/2024/Short_Papers — MIDL 2024 Short Papers_

### Official Review · Reviewer_9hAC · 2024-04-25

**Confidence:** 4
**Final Rating:** 3.5

**Review:**

The paper introduces some novel insight for the camlyon16 dataset. The paper can be improved by addressing the following questions:

1. How do instance-based MIL methods compare to embedding-based MIL methods in terms of performance and complexity?
2.	What are some key findings regarding the impact of SSL feature extractors on MIL performance in digital pathology?
3.	How does the MixMIL method with ResNet50 backbone pre-trained with DINO stand out in the results on the Camelyon16 dataset?

---

### Decision · Program_Chairs · 2024-04-26

Accept